# Clinical and Biological Predictors of Cancer Incidence and Mortality in Patients with Stable Coronary Artery Disease

**DOI:** 10.3390/ijms241311091

**Published:** 2023-07-04

**Authors:** Jonica Campolo, Andrea Borghini, Marina Parolini, Antonella Mercuri, Stefano Turchi, Maria Grazia Andreassi

**Affiliations:** 1CNR Institute of Clinical Physiology, ASST Grande Ospedale Metropolitano Niguarda, 20162 Milan, Italy; marina.parolini@cnr.it; 2CNR Institute of Clinical Physiology, 56124 Pisa, Italy; andrea.borghini@cnr.it (A.B.); antonella.mercuri@cnr.it (A.M.); stefano.turchi@cnr.it (S.T.); mariagrazia.andreassi@cnr.it (M.G.A.)

**Keywords:** cancer, coronary artery disease, risk factors, circulating biomarkers

## Abstract

Clinical and epidemiological evidence has recently revealed a link between coronary artery disease (CAD) and cancer. Shared risk factors and common biological pathways are probably involved in both pathological conditions. The aim of this paper was to evaluate whether and which conventional risk factors and novel circulating biomarkers could predict cancer incidence and death in patients with CAD. The study included 750 CAD patients, who underwent blood sampling for the evaluation of systemic inflammatory indexes (NLR and SII) and specific biomarkers of oxidative damage (leukocyte telomere length (LTL), mitochondrial DNA copy number (mtDNAcn)). Study participants were followed up for a mean of 5.4 ± 1.2 years. Sixty-seven patients (8.9%) developed cancer during the follow-up time, and nineteen (2.5%) died of cancer. Cox multivariable analysis revealed that age (HR = 1.071; 95% CI: 1.034–1.109; *p* < 0.001), smoking habit (HR = 1.994; 95% CI: 1.140–3.488; *p* = 0.016), obesity (HR = 1.708; 95% CI: 1.022–2.854; *p* = 0.041) and SII (HR = 1.002; 95% CI: 1.001–1.003; *p* = 0.045) were associated with cancer incidence, while only age (HR = 1.132; 95% CI: 1.052–1.219; *p* = 0.001) was a predictor of cancer death. Patients with lung and gastrointestinal cancers had significantly higher median mtDNAcn levels than those without cancer. Our study suggests that aggressive risk factor modification and suppression of chronic inflammation may be essential to preventing cancer in CAD patients.

## 1. Introduction

Recent clinical and epidemiological evidence supports the hypothesis of a bidirectional association between cardiovascular diseases and cancer [1,2]. Patients diagnosed with cancer are more susceptible to developing cardiovascular diseases than healthy subjects [3]. Similarly, patients with coronary artery disease (CAD) and heart failure (HF) have a higher cancer risk than the general or non-affected population [4,5]. 

CAD is an inflammatory condition of the arterial wall characterized by atherosclerotic plaque formation in the epicardial arteries that can be either obstructive or non-obstructive. Major coronary risk factors include elevated plasma cholesterol level, smoking, hypertension, diabetes mellitus and obesity. The clinical manifestations of CAD include chronic stable angina, defined by exercise-related chest pain. Patients with stable angina may develop acute coronary events, such as unstable angina and acute myocardial infarction.

Moreover, subjects who have developed myocardial infarction (MI) have higher short- and long-term hazard ratios for cancer than those without MI [4], and patients with atherosclerotic cardiovascular diseases exhibit two-fold higher incidence of cancer development than those with non-atherosclerotic CVD [6]. Moreover, coronary microvascular dysfunction is associated with cancer incidence in patients presenting with non-obstructive coronary artery disease [7]. CAD seems to be associated particularly with colorectal and lung cancers [1].

Although it is well known that cancer patients might also develop cardiovascular diseases as a result of chemotherapy and radiotherapy cardiotoxicity [8], less is known about the cause of the increased cancer incidence and mortality in CAD and HF. However, the potential of CAD as a causal factor in cancer remains unknown, and often, the findings are inconsistent and contradictory [9]. 

Patients with mild CAD prior to cancer diagnosis may have progression in disease severity due to the pro-inflammatory state resulting from the tumor itself [10].

Shared risk factors such as age, smoking, hypertension, obesity and diabetes mellitus could promote the pathogenesis of both diseases through a common perturbed milieu in which chronic inflammation and oxidative stress play an important role [2,11]. 

Indeed, oxidative stress conditions due to inadequate antioxidant defenses and/or reactive oxygen species overproduction are the main key triggers of inflammation in all aspects of coronary disease and acute thrombotic events, including endothelial dysfunction, the oxidation of LDL leading to atheroma formation, plaque rupture and recurrent thrombosis [2,11].

Importantly, oxidative and inflammation-based prognostic scores, such as neutrophil-to-lymphocyte ratio (NLR) or systemic immune-inflammation index (SII) have been recently introduced as markers of systemic inflammatory response and for prognosis in both cancer and cardiovascular diseases [12,13,14]. Additionally, intensive medical ionizing radiation exposure [15,16,17] and drugs [18,19] in diagnosis and treatment may be iatrogenic risk factors for developing cancer in patients with CAD.

Furthermore, cancer and atherosclerosis also share plausible biological and genetic mechanisms that are poorly understood. For instance, changes in leukocyte telomere length (LTL) and mitochondrial DNA copy number (mtDNAcn) in blood, which are the expression of genetic instability and DNA damage, are considered other important biological mechanisms initiating and contributing to not only cancer risk [20,21] but also vascular diseases and atherogenesis [22]. The presence of shortened telomeres and mitochondrial DNA damage is associated with different types of human cancer [1] and has been verified in human plaques [23] and peripheral cells of atherosclerotic patients [24].

However, the prognostic effect of indexes of systemic inflammation and DNA-based biomarkers, as well as the impact of clinical risk factors on cancer incidence and mortality, remains largely unexplored in CAD patients. Accordingly, we aim to evaluate whether and which conventional risk factors and novel circulating biomarkers could predict cancer incidence and death in patients with CAD.

## 2. Results

### 2.1. Demographic and Clinical Characteristics

The present study included 750 CAD patients (653 males and 97 females) with a median age of 66 [60–71 quartiles] years. Hypertension was present in 48%; dyslipidemia, in 69%; diabetes, in 15%; and smoking habit, in 62%. A total of 29% had a diagnosis of obesity; 55%, a family history of CAD; and 52%, a previous acute MI (AMI).

### 2.2. Follow-Up and Cancer Outcome

Study participants were followed up for a mean of 5.4 ± 1.2 years. Sixty-seven patients (8.9%) developed cancer during follow-up time. The most common cancers included prostate cancer (28.4%), lung cancer (16.4%), gastrointestinal and bladder cancers (10.4%), breast and kidney cancers (6.0%), and brain cancer and leukemia (4.5%). The other types of cancer had a frequency of ≤3%, as reported in Table 1.

Nineteen patients (2.5%) died of cancer; the main cause of death was lung tumor (26.3%).

### 2.3. Association between Risk Factors for CAD and Incidence of Cancer

The baseline clinical and biochemical characteristics of patients with and without cancer events are depicted in Table 2. We performed univariable Cox regression analysis to investigate the association between the most important risk factors and the incidence of cancer. Age, smoking habit, obesity and a number of revascularization procedures ≥2 were associated with the event. Multivariable Cox regression analysis adjusted for the circulating biomarkers showed that age (HR = 1.071; 95% CI: 1.034–1.109; *p* < 0.001), smoking habit (HR = 1.994; 95% CI: 1.140–3.488; *p* = 0.016), obesity (HR = 1.708; 95% CI: 1.022–2.854; *p* = 0.041) and SII (HR = 1.002; 95% CI: 1.001–1.003; *p* = 0.045) were significantly associated with cancer incidence (Table 3). 

Figure 1 shows the adjusted event-free survival curves according to smoking (panel A) and obesity (panel B), respectively.

### 2.4. Association between Risk Factors for CAD and Cancer Death

The clinical and biochemical parameters of patients with or without cancer death are represented in Table 4. Univariable Cox regression analysis showed that age, diabetes, obesity and a number of revascularization procedures ≥2 were related with cancer mortality. The multivariable Cox regression model, adjusted for the circulating biomarkers, indicated that only age (HR = 1.132; 95% CI: 1.052–1.219; *p* = 0.001) was a significant predictor of cancer death (Table 5).

### 2.5. Circulating Biomarkers Levels in Patients with Different Types of Cancer vs. Those without Cancer

We compared the levels of circulatig biomarkers in patients with lung and gastrointestinal cancers, the most relevant types of neoplasia associated with CAD, and those of patients without cancer (WC). Furthermore, we performed the same comparison between patients with prostate cancer, which is the prevalent type of neoplasia found in our population (28.4%), and WC patients. 

We found that patients with lung + gastrointestinal cancers had significantly higher median mtDNAcn levels than the WC group (42.5 [29.1–62.3] vs. 29.3 [17.6–45.1], *p* = 0.014), as depicted in Figure 2, panel A, while the median mtDNAcn value of patients with prostate neoplasia did not differ from that of WC patients, as shown in Figure 2, panel B. No other circulating biomarkers were statistically different between groups.

## 3. Discussion

This is the first molecular clinical study to simultaneously investigate the prognostic effect of clinical factors and biological markers on cancer prevalence and mortality among stable CAD patients. 

In recent years, an increasing number of epidemiological and clinical studies have reported that patients with cardiovascular diseases are at higher risk of developing cancer [1,2,3,4,5,6,7]. A very recent meta-analysis revealed an increased risk of incident cancer after a CAD event, particularly for lung and colorectal cancers, suggesting that detailed cancer surveillance programs should be implemented in patients with CAD to reduce cancer-related morbidity and mortality [1]. However, more research is warranted regarding the causes and risk factors of cancer in these patients [1]. 

Nowadays, it is well established that atherosclerosis and cancer share similar epidemiological risk factors and some common basic molecular pathways [11,25,26]. Chronic low-grade systemic inflammation has been proposed as the major unifying pathophysiological process in the pathogenesis of both diseases [11,25,26]. Indeed, chronic inflammation occurs in the presence of common clinical conditions, such hypertension, diabetes, dyslipidemia and obesity. Accordingly, our findings show that some traditional cardiovascular risk factors, such as age, smoking and obesity, explain a substantial fraction of the association between CAD, and both cancer incidence and mortality in our population. Specifically, chronologic age was independently associated with both incidence of and mortality from cancer in our CAD population, confirming that cancer is predominantly a disease of ageing. 

Furthermore, our results confirm that smoking and obesity are preventable and well-known risk factors for multiple types of cancer [27,28]. Other sources of inflammation include exposure to environmental toxicants and ionizing radiation [29]. Importantly, a previous study has shown that exposure to low-dose ionizing radiation from cardiac imaging and therapeutic procedures after acute myocardial infarction is associated with an increased risk of cancer [15]. In our study, increased cancer risks were also shown for patients with >2 catheterization procedures; this increased hazard risk, however, was not statistically significant (*p* = 0.096).

Additionally, the recently developed systemic immuno-inflammation index (SII) was found to be an independent risk indicator for cancer incidence, supporting the idea that chronic inflammation may play a considerable role in cancer development in CAD patients. The SII is a relatively novel systemic inflammatory marker based on lymphocyte, neutrophil and platelet counts that can provide a relatively complete picture of the balance between the inflammatory and immune states of the organism [30].

Consistently with our findings, several studies reported that the SII is a strong and independent predictor for incident cancer development in healthy individuals [31], as well as a prognostic factor for solid cancers, such as colorectal cancer, hepatocellular carcinoma and pancreatic cancer [32,33]. However, further understanding is needed to define the role of inflammation-based biomarkers in cancer risk in CAD patients in order to better guide prevention and early diagnosis.

Regarding common molecular pathways, damage of nuclear and mitochondrial DNA may be the most crucial mechanism in the etiology and progression of both atherosclerosis and cancer [20,21,22,23,24]. Indeed, both mitochondria and telomeres are critical regulators of cellular aging, and their dysfunction may increase the risk of age-related diseases, including atherosclerosis and cancer.

Accordingly, mtDNAcn and TL have been investigated as potential DNA-based biomarkers of CAD and various types of cancer [20,21,22,23,24]. Of note, the joint measurement of LTL and mtDNAcn has also been suggested to increase their predictive value for cancer risk [34,35].

In our study, we detected no statistically significant effect of LTL and mt-DNA on the risk of overall cancer prevalence and mortality, in line with recent evidence in the general population [35]. Furthermore, Li and colleagues showed that both mtDNAcn and telomere length are associated with the prevalence as well as future risk of cancer but in a cancer-specific manner, suggesting that the application of these biomarkers for evaluating cancer risk may be premature in this stage [34]. However, mtDNAcn levels increased in patients who developed lung and gastrointestinal cancers compared with those without cancer in our population.

Altered mtDNAcn is an emerging biomarker of several types of cancer [36]. Significant associations between mtDNAcn, and lung and colorectal cancer diagnosis and prognosis have been reported by several studies [37,38,39], although the direction of the association varied. Elevated leukocyte mtDNAcn was associated with decreased cancer survival in some cases [39], while in others, high mtDNAcn in leukocytes or in tumor tissue identified a lower risk of death [40,41]. Current knowledge of mtDNAcn regulation in cancer is still unclear and needs to be investigated in a larger cohort.

Some study limitations should be acknowledged. First, we had a relatively limited sample size. This forced us to combine overall cancer, preventing the possibility to analyze incidence of and mortality from specific cancer forms. Second, we cannot completely rule out the influence of cardiovascular treatment on the biological measurements as well as on the clinical endpoint. Finally, we cannot exclude the possibility of residuals due to unmeasured confounders or measurement error. In conclusion, several known risk factors and an activated inflammatory profile were associated with increased incidence of cancer in our cohort. Instead, we did not find any association between cancer incidence and mortality, and markers of DNA damage in the CAD population; however, mtDNAcn levels were higher in CAD patients with pulmonary and gastrointestinal neoplasias than in those without cancer, indicating a role of mitochondrial dysfunction in their pathogenesis.

This study suggests that aggressive risk factor modification and suppression of chronic inflammation may be essential to preventing cancer in CAD patients.

Further identification of cancer risk factors in a larger CAD population could be helpful in developing new effective preventive strategies to reduce future morbidity.

## 4. Materials and Methods

### 4.1. Study Population

This retrospective cohort study included adult patients with stable CAD, defined angiographically as having ≥1 major coronary vessel with at least 50% stenosis. The study population was recruited within the framework of the Italian cohort GENOCOR (Genetic Mapping for Assessment of Cardiovascular Risk; ClinicalTrials.gov Identifier: NCT01506999; https://clinicaltrials.gov/ct2/show/NCT01506999, accessed on 10 January 2012). All patients underwent blood sampling at baseline for the evaluation of routine biochemical parameters and specific biomarkers of oxidative damage (leukocyte telomere length (LTL), mitochondrial DNA copy number (mtDNAcn)). Study participants were followed up for a mean of 5.4 ± 1.2 years. None of them had a cancer diagnosis at the time of blood sampling.

Data regarding demographic and clinical history were collected for all patients. Cardiovascular risk factors were defined according to current guidelines for arterial hypertension (systolic and diastolic blood pressure ≥ 140 and 90 mmHg, respectively, or on antihypertensive medications), diabetes (fasting plasma glucose > 120 mg/dL), obesity (body mass index > 30 kg/m^2^) and dyslipidemia (LDL ≥ 160 mg/dL or under treatment with lipid-lowering medications). Smokers were classified as individuals who smoked at least 3 cigarettes per day at the time of analysis; past smokers had quit smoking at least 6 months prior; and non-smokers were individuals who had never smoked. Smoking patients were the combined groups of past and current smokers.

The study protocol was approved by the Institution’s Ethics Committee, and all patients gave written informed consent.

### 4.2. Study Design

In the first analysis, the outcome was incidence of cancer, including fatal and non-fatal. The time of study entry was considered the time of blood collection, and the end of follow-up time was the date of cancer diagnosis, death from any other causes or the last visit, whichever came first. Patients who died of other causes during follow-up were censored. 

The outcome of the second analysis was mortality for cancer.

### 4.3. Routine and Specific Biochemical Measurements

Blood samples were drawn from all patients under fasting conditions and processed according to standard operating procedures for various biochemical parameters using routine automated laboratory analyzers and standard methods.

Briefly, lipid profile (total cholesterol, LDL cholesterol, HDL cholesterol and triglycerides) and glucose were measured using a Synchron CX analyzer (Beckman Systems, Fullerton, CA, USA). LDL cholesterol concentration was calculated using the Friedewald equation. Quantitative glycated haemoglobin concentration was measured using the DiaSTAT haemoglobin A1C programme on the BioRad DiaSTAT analyzer (Bio-Rad, Hercules, CA, USA), following the standard recommended procedures.

Total and differential leukocyte counts (neutrophils, eosinophils, basophils, lymphocytes and monocytes) were determined using an automated Coulter counter (model MAXM; Instrumentation Laboratory, Miami, FL, USA).

Two different indexes were used to evaluate the degree of systemic inflammation: the neutrophil/lymphocyte ratio (NLR) and the platelets × neutrophils/lymphocytes, 10^9^/L (SII) [42]. The NLR is a marker of subclinical inflammation, with a higher value in exacerbations, in many lung and cardiac diseases and in obesity [42,43]. The SII is a novel immune-inflammation index that has high prognostic value in cancer patients [42,44]. 

Genomic DNA was extracted from peripheral blood leukocytes using a QIAGEN BioRobot^®^ EZ1 System. LTL and mtDNAcn were measured using quantitative real-time methods (CFX384 Touch Real-time PCR detection system; Bio-Rad, Hercules, CA, USA) following previously described protocols [45,46]. Briefly, LTL was measured in genomic DNA by determining the ratio of telomere repeat copy number (T) to single-copy gene (S) copy number (T/S ratio). The relative telomere length was calculated with the following equation: T/S ratio = 2^−ΔΔCt^, where Ct is the cycle threshold and ΔCt = Ct × telomere − Ct × single-copy gene. The T/S ratio reflects the average length of telomeres across all leukocytes [45].

As regards to mtDNAcn, the ND1 gene in the undeleted region for the reference sequence of mtDNA as an internal control (mtND1) and the human ß-globin gene of genomic DNA (g-DNA) were amplified using PCR on both g-DNA and mtDNA. All samples were run in duplicate, and only the average Ct values were determined. ΔCt values were computed as the difference between the Ct for the ß-globin gene and the Ct for the NDI1 gene and used for the measurement of mtDNAcn relative to gDNA. mtDNAcn was calculated using the 2^−ΔCt^ method (ΔCt = Ct mtND1 − CtgDNA) [46].

### 4.4. Statistical Analysis

Data are presented as medians, and I and III quartiles [I–III quartiles] for continuous variables, and as numbers (percentages) for categorical variables.

We used univariable and multivariable Cox proportional hazards regression analyses to estimate hazard ratios (HRs) and their 95% confidence intervals (CIs) for the associations of covariates and cancer events (incidence of cancer), and of covariates and cancer death. All the variables with a *p*-value < 0.20 in the univariable analysis entered in the multivariable model. In addition, multivariable models were adjusted for the circulating biomarkers of systemic inflammation (NRL, SII) and oxidative stress (LTL, mtDNAcn). Adjusted survival curves were plotted from the multivariable Cox regression model.

Skewed data were compared between groups using the Mann–Whitney U test.

Statistical analyses were performed using the SPSS ver. 24.0 software package (IBM SPSS, New York, NY, USA). A *p*-value < 0.05 was considered statistically significant.

## Figures and Tables

**Figure 1 ijms-24-11091-f001:**
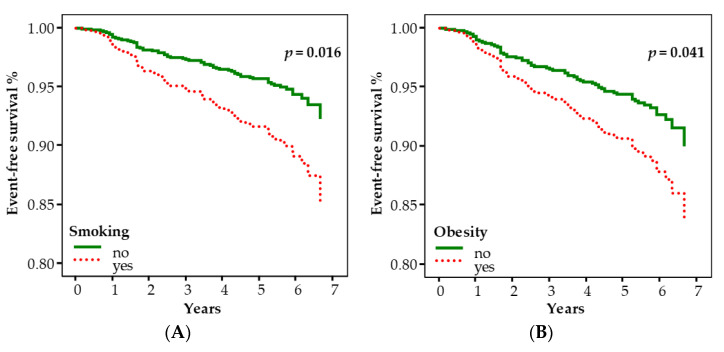
The event-free survival curves according to smoking (**A**) and obesity (**B**), adjusted with the multivariable Cox model.

**Figure 2 ijms-24-11091-f002:**
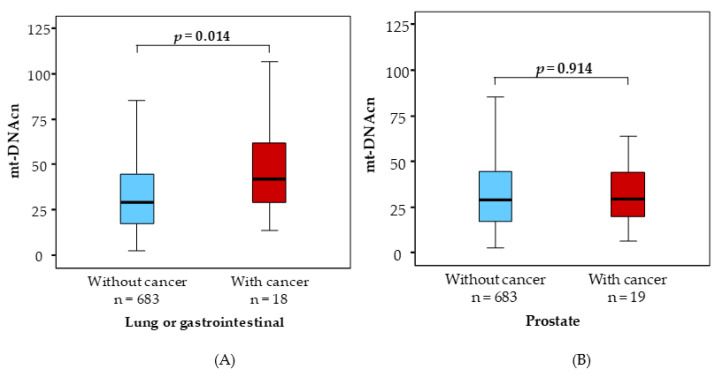
Mitochondrial DNA copy number comparison between patients with different types of cancer (Lung or gastrointestinal, panel **A** and Prostate, panel **B**) and those without neoplasia.

**Table 1 ijms-24-11091-t001:** Site of cancer and frequency.

Type	Number	%
Prostate	19	28.4
Lung	11	16.4
Gastrointestinal	7	10.4
Bladder	7	10.4
Breast	4	6.0
Kidney	4	6.0
Brain	3	4.5
Leukemia	3	4.5
Mouth	2	3.0
Skin	2	3.0
Thyroid	2	3.0
Ovarian	1	1.5
Myeloma	1	1.5
Mediastinal	1	1.5

**Table 2 ijms-24-11091-t002:** Baseline characteristics and univariable Cox regression analysis for cancer incidence.

	Cancer Event		HR	95% CI	*p*-Value
	Yes (*n* = 67)	No (*n* = 683)			
Age, years	68 [63–72]	65 [59–70]	1.053	1.020–1.087	0.001
Male gender, *n* (%)	56 (84)	597 (87)	0.720	0.377–1.376	0.320
Cardiovascular risk factors					
Family history, *n* (%)	33 (49)	380 (56)	0.772	0.478–1.246	0.290
Smoking habit, *n* (%)	47 (70)	414 (61)	1.476	0.874–2.491	0.145
Diabetes, *n* (%)	10 (15)	101 (15)	1.050	0.536–2.057	0.887
Hypertension, *n* (%)	28 (42)	333 (49)	0.770	0.474–1.252	0.292
Dysplipidemia, *n* (%)	49 (73)	469 (69)	1.205	0.702–2.068	0.499
Obesity, *n* (%)	24 (36)	194 (28)	1.397	0.847–2.302	0.190
Previous AMI, *n* (%)	36 (52)	354 (54)	0.881	0.538–1.444	0.616
Number of revascularization procedures					
<2, *n* (%)	52 (78)	581 (85)	Reference		
≥2, *n* (%)	15 (22)	102 (15)	1.628	0.916–2.894	0.096
Circulating biomarkers					
LTL	1.01 [0.67–1.29]	1.00 [0.70–1.40]	0.952	0.643–1.409	0.806
mtDNAcn	33 [19–49]	29 [18–45]	1.000	0.994–1.005	0.871
NLR	2.10 [1.63–2.70]	2.10 [1.61–2.77]	0.927	0.739–1.163	0.512
SII, 10^9^/L	521 [376–699]	483 [353–642]	1.000	0.999–1.001	0.896

AMI: acute myocardial infarction; LTL: leukocyte telomere length; mt: mitochondrial; cn: copy number; NLR: neutrophil-to-lymphocyte ratio; SII: systemic immune-inflammation index; HR: hazard ratios; CI: confidence intervals.

**Table 3 ijms-24-11091-t003:** Multivariable Cox regression analysis for cancer incidence.

	HR	95% CI	*p*-Value
Age	1.071	1.034–1.109	<0.001
Smoking habit	1.994	1.140–3.488	0.016
Obesity	1.708	1.022–2.854	0.041
No. of revascularization procedures ≥ 2	1.658	0.912–3.017	0.098
LTL	0.893	0.586–1.361	0.598
mtDNAcn	1.009	0.995–1.009	0.653
NLR	0.621	0.379–1.018	0.059
SII	1.002	1.001–1003	0.045

LTL: leukocyte telomere length; mt: mitochondrial; cn: copy number; NLR: neutrophil-to-lymphocyte ratio; SII: systemic immune-inflammation index; HR: hazard ratios; CI: confidence intervals.

**Table 4 ijms-24-11091-t004:** Baseline characteristics and univariable Cox regression analysis for cancer death.

	Cancer Death		HR	95% CI	*p*-Value
	Yes (*n* = 19)	No (*n* = 731)			
Age, years	70 [67–75]	65 [59–70]	1.122	1.049–1.200	0.001
Male gender	16 (84)	637 (87)	0.738	0.214–2.539	0.630
Cardiovascular risk factors					
Family history, *n* (%)	9 (47)	404 (55)	0.733	0.298–1.804	0.499
Smoking, *n* (%)	12 (63)	449 (61)	1.064	0.419–2.702	0.897
Diabetes, *n* (%)	5 (26)	106 (15)	2.125	0.765–5.902	0.148
Hypertension, *n* (%)	10 (53)	351 (48)	1.228	0.499–3.021	0.656
Dysplipidemia, *n* (%)	12 (63)	506 (69)	0.751	0.296–1.909	0.548
Obesity, *n* (%)	8 (42)	210 (29)	1.824	0.734–4.537	0.196
Previous AMI, *n* (%)	13 (68)	377 (52)	1.672	0.622–4.497	0.308
Number of revascularization procedures					
<2, *n* (%)	14 (74)	619 (85)	Reference		
≥2, *n* (%)	5 (26)	112 (15)	1.964	0.707–5.455	0.195
Circulating biomarkers					
LTL	1.04 [0.67–1.70]	1.00 [0.70–1.39]	1.402	0.759–2.587	0.280
mtDNAcn	30 [17–59]	29 [18–45]	0.999	0.987–1.011	0.878
NLR	2.20 [1.54–3.17]	2.10 [1.61–2.77]	1.020	0.729–1.429	0.907
SII, 10^9^/L	493 [322–765]	485 [356–649]	1.000	0.998–1.001	0.762

AMI: acute myocardial infarction; LTL: leukocyte telomere length; mt: mitochondrial; cn: copy number; NLR: neutrophil-to-lymphocyte ratio; SII: systemic immune-inflammation index; HR: hazard ratios; CI: confidence intervals.

**Table 5 ijms-24-11091-t005:** Multivariable Cox regression analysis for cancer death.

	HR	95% CI	*p*-Value
Age	1.132	1.052–1.219	0.001
Diabetes	1.573	0.524–4.725	0.420
Obesity	2.296	0.856–6.155	0.099
No. of revascularization procedures ≥ 2	2.271	0.779–6.617	0.133
LTL	1.081	0.549–2.128	0.882
mtDNAcn	1.003	0.989–1.017	0.644
NLR	1.205	0.531–2.736	0.656
SII	0.999	0.995–1.002	0.534

LTL: leukocyte telomere length; mt: mitochondrial; cn: copy number; NLR: neutrophil-to-lymphocyte ratio; SII: systemic immune-inflammation index; HR: hazard ratios; CI: confidence intervals.

## Data Availability

The data presented in this study are available upon request from Maria Grazia Andreassi and are available to researchers of academic institutes who meet the criteria for access to confidential data.

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
