# Peer review of "Clinical and Biological Predictors of Cancer Incidence and Mortality in Patients with Stable Coronary Artery Disease"

_ijms, 2023, doi:10.3390/ijms241311091_

Round 1

Reviewer 1 Report (Previous Reviewer 1)

I would like to thank the authors for addressing my initial comments. The authors have very effectively addressed all comments. Following the revision to the article, I feel that this manuscript is now acceptable for publication.

Reviewer 2 Report (Previous Reviewer 2)

Dear Campolo et al.,

Thank you for submitting your manuscript entitled “Clinical and biological predictors of cancer incidence and mortality in patients with stable coronary artery disease” to “International Journal of Molecular Sciences”. The study examined new circulating biomarkers and risk factors for cancer in patients with coronary artery disease that predict cancer incidence and mortality.

The large number of CAD patients in the sample, the lengthy follow-up time, and the clearly defined scientific methodology are this observational study's key advantages. This intriguing study emphasizes the significance of risk factor reduction and chronic inflammation regulation to shield CAD patients from malignancy.

I appreciate everyone's efforts in creating such a fantastic work.

This manuscript is a resubmission of an earlier submission. The following is a list of the peer review reports and author responses from that submission.

Round 1

Reviewer 1 Report

Coronary artery disease (CAD) is the most common type of heart disease. CAD happens due to the build-up of cholesterol and other material, called plaque, on their inner walls. CAD and cancer appear to share many common risk factors. The aim of this paper was to evaluate whether and which conventional risk factors and novel circulating biomarkers could predict cancer incidence and death in patients with CAD. The current study suggests that aggressive risk factor modification and suppression of chronic inflammation may be essential for preventing cancer in CAD patients.

1.      Discuss the cause of coronary artery disease and its link with cholesterol deposition in arteries.

2.      Briefly introduce the types of CAD in the introduction section

3.      Oxidative stress, hyperglycemia, hypertriglyceridemia, and low AOP contribute to oxidation of LDL and atherosclerosis. Elevated triglycerides increase oxidizability of LDL while HDL reduces oxidizability. The oxidation of LDL induces accumulation of LDL cholesterol and an inflammatory response.

In light of the aforementioned information, it is necessary to briefly correlate CAD with oxidative stress.

4.      DNA damage caused by point mutation, deletion, insertion, recombination, rearrangements, and amplifications, as well as chromosomal aberrations can lead to cancer initiation. Please, cite the following article.

Natural products: implication in cancer prevention and treatment through modulating various biological activities. https://doi.org/10.2174/1871520620666200705220307.

5.      Provide the URL link of Italian cohort 205 GENOCOR (Genetic Mapping for Assessment of Cardiovascular Risk, ClinicalTrials.gov 206 Identifier: NCT01506999).

6.      Describe the methodology to quantify total cholesterol, LDL, VLDL and hematological parameters including total leucocyte counts, differential leucocyte counts such as neutrophils, lymphocytes, and monocytes. If, the biochemical investigations were carried out using commercially available kits according to the manufacturer instructions, then provide the kit number with its manufacturer name.

7.      Two different indexes were used to evaluate the degree of systemic inflammation: the neutrophils/lynphocytes ratio (NLR) and the platelets x neutrophils/lymphocytes, 109/L (SII).

Provide a strong reference for the formula used above and discuss the index  determining the inflammation.

8.      LTL was measured in genomic DNA by determining the T/S ratio. A relative telomere length was calculated by the equation: T/S ratio = 2–ΔΔCt, where Ct is a threshold cycle and ΔCt = ct × telomere − Ct × single copy gene. Explain the method to calculate T and S.

9.      The methodology is based on real time PCR but the authors did not included the necessity of calculation of ct value in the manuscript. Include the significance of ct value and mention following sentences in section 4.3.

In a real time PCR assay a positive reaction is detected by accumulation of a fluorescent signal. The CT (cycle threshold) is defined as the number of cycles required for the fluorescent signal to exceed background levels. CT levels are inversely proportional to the amount of target nucleic acid in the sample (i.e., the lower the CT level the greater the amount of target nucleic acid in the sample). Values < or = 36 are positive. Results of suspect, inconclusive or weak positive with Ct values from 37-40 indicate minimal amounts of target nucleic acid which could represent early or late infection, residual vaccine or environmental contamination.

I think that mentioning these sentences in brief in the discussion section could be very beneficial.

10.     Malondialdehyde level is commonly known as a marker of oxidative stress and the antioxidant status in cancerous patients. I suggest to determine MDA level in blood of patients.

11.   The conclusions are too few compared to the extensive discussion in the text. I think they should be expanded and future prospective should be included.

12.  The authors have not provided the significance of this study. In which particular aspect, this study is different from the rest published so far.

13.  Abbreviations need an explanation.

14.  Some introductory lines should be written in the beginning of results section to better linkage the different typologies of results.

15.  The English language is adequate, It needs minor editing.

The English language is adequate, It needs minor editing.

Author Response

Prof. Dr. Andreea Margareta Sinka

Assigned Editor of International Journal of Molecular Sciences

Manuscript Number: Ref. ID ijms-2424568

Title: CLINICAL AND BIOLOGICAL PREDICTORS OF CANCER INCIDENCE AND MORTALITY IN PATIENTS WITH STABLE CORONARY ARTERY DISEASE

Special Issue: Advances in Molecular Pathophysiology of Cardiovascular Diseases

Dear Prof. Sinka,  

please find attached the revised version of the above referenced manuscript.

We are grateful to the Reviewers for their useful remarks and suggestions, that have indeed helped us to improve our work.

We tried to address all the constructive and helpful comments raised by the Reviewers.

Below the Reviewers’ comments (in italic text) and our answers (in plain text).

Changes have been marked up using the “Track Changes” function in MS Word/LaTeX.

We hope that the manuscript is suitable for publication in its present form.

Yours sincerely,

Dr. Jonica Campolo

The authors thank the Reviewer 1 for his/her through review and salient observations.

Reviewers' comments:

Reviewer #1

Coronary artery disease (CAD) is the most common type of heart disease. CAD happens due to the build-up of cholesterol and other material, called plaque, on their inner walls. CAD and cancer appear to share many common risk factors. The aim of this paper was to evaluate whether and which conventional risk factors and novel circulating biomarkers could predict cancer incidence and death in patients with CAD. The current study suggests that aggressive risk factor modification and suppression of chronic inflammation may be essential for preventing cancer in CAD patients.

Point 1. Discuss the cause of coronary artery disease and its link with cholesterol deposition in arteries.

Point 2. Briefly introduce the types of CAD in the introduction section

As suggested by reviewer, we briefly discussed these two points in the “Introduction” from line 34 to line 46. We also added a new reference (number 6, Suzuki et al.)

Point 3. Oxidative stress, hyperglycemia, hypertriglyceridemia, and low AOP contribute to oxidation of LDL and atherosclerosis. Elevated triglycerides increase oxidizability of LDL while HDL reduces oxidizability. The oxidation of LDL induces accumulation of LDL cholesterol and an inflammatory response.

In light of the aforementioned information, it is necessary to briefly correlate CAD with oxidative stress.

We briefly discussed the point from line 58 to line 63. Anyway, the paper is a research investigation which does not aim to discuss in detail a complex topic as the pathogenesis of coronary artery disease and the causes, including the link with inflammation, oxidative stress, hyperglycemia, hypertriglyceridemia, and low antioxidant potential.

Point 4. DNA damage caused by point mutation, deletion, insertion, recombination, rearrangements, and amplifications, as well as chromosomal aberrations can lead to cancer initiation. Please, cite the following article.

Natural products: implication in cancer prevention and treatment through modulating various biological activities. https://doi.org/10.2174/1871520620666200705220307.

We thank a lot the reviewer for the suggestion. Unfortunately, we have not been able to read this manuscript in its entirety because it is not open access. However, from the information contained in the abstract, we think it may be confusing and not relevant to our topic.

Point 5. Provide the URL link of Italian cohort 205 GENOCOR (Genetic Mapping for Assessment of Cardiovascular Risk, ClinicalTrials.gov 206 Identifier: NCT01506999).

As the reviewer advised, we added the link of GENOCOR In “Material and Methods” line 239.

Point 6. Describe the methodology to quantify total cholesterol, LDL, VLDL and hematological parameters including total leucocyte counts, differential leucocyte counts such as neutrophils, lymphocytes, and monocytes. If, the biochemical investigations were carried out using commercially available kits according to the manufacturer instructions, then provide the kit number with its manufacturer name.

We better specified that blood samples were immediately assessed by routine automated laboratory analyzers and standard method in the biochemical and hematological laboratories (from line 264 to line 271).

Point 7. Two different indexes were used to evaluate the degree of systemic inflammation: the neutrophils/lynphocytes ratio (NLR) and the platelets x neutrophils/lymphocytes, 109/L (SII).

Provide a strong reference for the formula used above and discuss the index determining the inflammation.

We provided references 42, 43 and 44 which contain NLR and SII formula and a brief explanation of inflammatory indexes meaning (from line 277 to line 279).

Point 8. LTL was measured in genomic DNA by determining the T/S ratio. A relative telomere length was calculated by the equation: T/S ratio = 2–ΔΔCt, where Ct is a threshold cycle and ΔCt = ct × telomere − Ct × single copy gene. Explain the method to calculate T and S.

We thank the reviewer for the specific indication. As you can see in “Material and Methods, Section 4.3, we added the definitions of T and C in the text (lines 284-285) and references on LTL [45] and mtDNAcn [46] measurements.

Point 9. The methodology is based on real time PCR but the authors did not included the necessity of calculation of ct value in the manuscript. Include the significance of ct value and mention following sentences in section 4.3. Ok ref 44 e 45

In a real time PCR assay a positive reaction is detected by accumulation of a fluorescent signal. The CT (cycle threshold) is defined as the number of cycles required for the fluorescent signal to exceed background levels. CT levels are inversely proportional to the amount of target nucleic acid in the sample (i.e., the lower the CT level the greater the amount of target nucleic acid in the sample). Values < or = 36 are positive. Results of suspect, inconclusive or weak positive with Ct values from 37-40 indicate minimal amounts of target nucleic acid which could represent early or late infection, residual vaccine or environmental contamination.

I think that mentioning these sentences in brief in the discussion section could be very beneficial.

Since its introduction in the early 1990s, Reverse transcription-polymerase chain reaction (RT-PCR) allows the sensitive, specific, and reproducible quantitation of the expression level of target genes. At the present, the RT-PCR has become one of the most widely used method in biomedical science research for semiquantitative analysis, and the basic principles of RT-PCR are well-known to researchers and clinicians. Consequently, we believe that mentioning the details on the quantitation of RT-PCR in the discussion is trivial and not pertinent.

Point 10. Malondialdehyde level is commonly known as a marker of oxidative stress and the antioxidant status in cancerous patients. I suggest to determine MDA level in blood of patients.

Unfortunately, we did not measure MDA concentrations.

Point 11. The conclusions are too few compared to the extensive discussion in the text. I think they should be expanded and future prospective should be included.

As the reviewer suggested, we expanded the “Conclusion” section (from line 222 to line 228). We included future prospective from line 231 to line 232.

Point 12. The authors have not provided the significance of this study. In which particular aspect, this study is different from the rest published so far.

To our knowledge, this is the first study to evaluate biomarkers of cancer risk, including leukocyte telomere length shortening, reduction of mitochondrial DNA copy number and oxidative and inflammation-based index in the setting of cancer incidence/mortality in patients with stable CAD. We highlighted this point at the beginning of Discussion (lines 152-154). In our opinion, our findings are important and timely to better define the association between CAD and incident cancer.

Point 13. Abbreviations need an explanation.

Thank you for your suggestion, we explained and added some abbreviations (line 41 “myocardial infarction”, MI; line 87 “acute MI”, AMI, Tables 2 and 4, lines 110 and 129).

Point 14. Some introductory lines should be written in the beginning of results section to better linkage the different typologies of results.

We added some introductory lines as suggested by the reviewer (lines 83 and 88-89).

Point 15. The English language is adequate, it needs minor editing.

We made minor editing in the revised version.

Reviewer 2 Report

Dear Campolo et al.,

Thank you for submitting your manuscript entitled “Clinical and biological predictors of cancer incidence and mor-2 tality in patients with stable coronary artery disease” to “International Journal of Molecular Sciences”. The presentation of abstract, introduction and discussion of the manuscript is impressive. Methods and results are pretty clear. A couple of minor suggestions:

1.     On line 32 of the introduction section, change “general population” to “healthy population”.  

2.     Please abbreviate the full meaning of “IMA” on line 67.

Thank you for considering my suggestions.

Author Response

Prof. Dr. Andreea Margareta Sinka

Assigned Editor of International Journal of Molecular Sciences

Manuscript Number: Ref. ID ijms-2424568

Title: CLINICAL AND BIOLOGICAL PREDICTORS OF CANCER INCIDENCE AND MORTALITY IN PATIENTS WITH STABLE CORONARY ARTERY DISEASE

Special Issue: Advances in Molecular Pathophysiology of Cardiovascular Diseases

Dear Prof. Sinka,  

please find attached the revised version of the above referenced manuscript.

We are grateful to the Reviewers for their useful remarks and suggestions, that have indeed helped us to improve our work.

We tried to address all the constructive and helpful comments raised by the Reviewers.

Below the Reviewers’ comments (in italic text) and our answers (in plain text).

Changes have been marked up using the “Track Changes” function in MS Word/LaTeX.

We hope that the manuscript is suitable for publication in its present form.

Yours sincerely,

Dr. Jonica Campolo

The authors thank the Reviewer 2 for the salient observations.

Reviewers' comments:

Reviewer #2  

Thank you for submitting your manuscript entitled “Clinical and biological predictors of cancer incidence and mor-2 tality in patients with stable coronary artery disease” to “International Journal of Molecular Sciences”. The presentation of abstract, introduction and discussion of the manuscript is impressive. Methods and results are pretty clear. A couple of minor suggestions:

The authors thank the reviewer for his/her kind words.

 Point 1. On line 32 of the introduction section, change “general population” to “healthy population”.

Thank you very much for your suggestion. Unfortunately, we cannot eliminate the sentence “general population” because the epidemiological research of Rinde et al., we cited (n. 4), is referring to large cohort recruited from a general population. However, in addition to general we introduced “non affected population” (see line 32).

Point 2. Please abbreviate the full meaning of “IMA” on line 67.

As you asked, we added the full meaning of IMA (line 87). Moreover, we changed IMA with AMI (lines 87, 110 and 129, Tables 2 and 4).

Reviewer 3 Report

The bidirectional relationship of interdependence between cardiovascular diseases and neoplasias is well known.

From this perspective, the topic addressed by the authors is particularly interesting, especially since the morbidity and mortality due to coronary diseases is increasing. 

The elements that significantly increase the scientific value of the work are:

- large number of CAD patients included in the study(750 CAD patients);

- prospective study, carried out over a long period of follow-up (5.4±1.2 years);

- another strong point of the study is the modified analysis in leukocytes telomere and mitochondrial DNA copy number, which are reduced in patients with neoplasia risk;

- the methodology approached is well defined.

Recommendations for authors:

- The chapter on the study methodology is usually found after the introduction, before the results.

- The study evaluates the simple or cumulative risk for the classic risk factors of developing a neoplasia or even death in the next 5 years, in patients with stable artery disease. So, in this research there is no evidence from which there is a greater risk of developing neoplasia in patients with stable coronary disease compared to those in the general population, although this is suggested in the introduction. I consider absolutely necessary the comparative analysis of the neoplasia risk in the studied group, with stable coronary disease, with a control group from the general population, in this sense the recommendation is to readjust the study methodology, expand the results and discussions.

The very title of this work "Clinical and biological predictors of cancer incidence and mortality in patients with stable coronary artery disease" requires a comparative approach of a group of patients with stable coronary disease, compared to a control group, from the general population.

Author Response

Prof. Dr. Andreea Margareta Sinka

Assigned Editor of International Journal of Molecular Sciences

Manuscript Number: Ref. ID ijms-2424568

Title: CLINICAL AND BIOLOGICAL PREDICTORS OF CANCER INCIDENCE AND MORTALITY IN PATIENTS WITH STABLE CORONARY ARTERY DISEASE

Special Issue: Advances in Molecular Pathophysiology of Cardiovascular Diseases

Dear Prof. Sinka,  

please find attached the revised version of the above referenced manuscript.

We are grateful to the Reviewers for their useful remarks and suggestions, that have indeed helped us to improve our work.

We tried to address all the constructive and helpful comments raised by the Reviewers.

Below the Reviewers’ comments (in italic text) and our answers (in plain text).

Changes have been marked up using the “Track Changes” function in MS Word/LaTeX.

We hope that the manuscript is suitable for publication in its present form.

Yours sincerely,

Dr. Jonica Campolo

Reviewers' comments:

Reviewer #3

The bidirectional relationship of interdependence between cardiovascular diseases and neoplasias is well known.

From this perspective, the topic addressed by the authors is particularly interesting, especially since the morbidity and mortality due to coronary diseases is increasing. 

The elements that significantly increase the scientific value of the work are:

- large number of CAD patients included in the study (750 CAD patients);

- prospective study, carried out over a long period of follow-up (5.4±1.2 years);

- another strong point of the study is the modified analysis in leukocytes telomere and mitochondrial DNA copy number, which are reduced in patients with neoplasia risk;

- the methodology approached is well defined.

We thank the reviewer for his/her appreciation.

Recommendations for authors:

Point 1. The chapter on the study methodology is usually found after the introduction, before the results.

Thanks for your hint, but the journal rules provide for this setting:  

  1. Introduction; 2. Results; 3. Discussion and Conclusion; 4. Material and Methods.

Point 2. The study evaluates the simple or cumulative risk for the classic risk factors of developing a neoplasia or even death in the next 5 years, in patients with stable artery disease. So, in this research there is no evidence from which there is a greater risk of developing neoplasia in patients with stable coronary disease compared to those in the general population, although this is suggested in the introduction. I consider absolutely necessary the comparative analysis of the neoplasia risk in the studied group, with stable coronary disease, with a control group from the general population, in this sense the recommendation is to readjust the study methodology, expand the results and discussions.

The very title of this work "Clinical and biological predictors of cancer incidence and mortality in patients with stable coronary artery disease" requires a comparative approach of a group of patients with stable coronary disease, compared to a control group, from the general population.

Our study is an observational study that was conducted to investigate the impact of clinical risk factors and novel circulating biomarkers on cancer in patients with CAD.   

In other words, we did not conduct a classical epidemiological study in order to investigate whether or not CAD is associated with an increased risk of developing cancer respect to the general population, comparing the observed incidence of cancer with the expected incidence calculated from national cancer rates by the standardized incidence ratio.

Anyway, we can tell you that the cancer incidence of 8.9% found in our CAD cohort is 13 times higher than those of general population (0,7%) estimated in the same range of age (30-74 years) between 2013 and 2015 period (https://www.ispro.toscana.it ).

However, we prefer to not discuss this data in the paper because any epidemiological study of cancer risk is challenging because they require a rigorous design, large numbers of patients and extended follow-up. 

On the other hand, an increasing number of epidemiological and clinical studies have recently reported that patients with cardiovascular diseases have a higher risk of developing cancer.

We believe that the strength of this paper includes, for the first time, the careful evaluation of clinical conditions and different cellular and molecular markers in order to gain insights on the risk factors (largely unexplored) of cancer in these patients. In our opinion, our findings are important and timely to better define the association between CAD and incident cancer. 

Round 2

Reviewer 3 Report

I keep my opinion that is absolutely necessary the comparative analysis of the neoplasia risk in the studied group with stable coronary disease, with a control group from the general population.

Without this analysis, the manuscript does not bring any novelty on the scientific level.